# Why Barlow Twins Work: The Critical Role of Normalization and Its Link to Sample Contrastive Learning

## Abstract

Barlow Twins is a feature-contrastive self-supervised learning framework built on the principle of redundancy reduction. The idea is to train a network by maximizing the correlation between corresponding features and minimizing the correlation between non-corresponding features in distorted views of the same image, through this facilitating effective pretraining of a backbone network for a subsequent classification head. This is achieved by diagonalizing the cross-correlation matrix of the network's representations and scaling it towards the identity matrix. We show that the cross-correlation matrix of distorted images is inherently symmetric, independent of the backbone network's weights, which leads to two key insights: (i) the cross-correlation matrix can always be diagonalized using a linear transformation (layer), and (ii) the core idea of maximizing correlations between corresponding features while minimizing them for non-corresponding features alone is insufficient for effective backbone network pretraining. Nevertheless, Barlow Twins provide highly effective pretraining. We show that this is due to the normalization of the cross-correlation matrix in the Barlow Twins cost function. This normalization leads to minima of the cost function which are equivalent to the minima of sample contrastive approaches to enforce invariance.

## 1 Introduction

In self-supervised learning, the goal is to learn meaningful representations without relying on labels, which can be costly to obtain, especially for large datasets. Approaches like SimCLR (Chen et al., 2020), SwAV (Caron et al., 2020), or SimSiam (Chen & Kaiming He, 2021) show that self-supervised methods can produce strong representations which achieve competitive results compared to supervised approaches. These approaches are often called sample-contrastive, as they make samplewise comparisons between different instances. Another class of self-supervised learning approaches is called feature-contrastive, which works by comparing different instances at the feature level rather than the sample level. Examples of these methods are the Barlow Twins (Zbontar et al., 2021), VICReg (Bardes et al., 2022), VICRegL (Bardes et al., 2024), or W-MSE (Ermolov et al., 2021). A significant advantage of all of these methods lies in their ability to harness the inherent structure of unannotated data.

The Barlow Twins framework (Zbontar et al., 2021) introduced feature-contrastive learning, grounded in the principle of redundancy reduction in neural representations, initially proposed by H. Barlow (Barlow et al., 1961). The Barlow Twins approach minimizes the distance between a modified cross-correlation matrix and the identity matrix in order to extract representations with decorrelated feature dimensions. In Zbontar et al. (2021), a ResNet-50 $f$ (He et al., 2016) is adapted by deleting the fully connected layer and applying a projector network, which is a large multilayer perceptron (MLP) $p$. In Zbontar et al. (2021), this MLP has three linear layers, each with 8192 output dimensions and no biases. The first two layers are followed by a batch normalization layer as well as a rectified linear unit (ReLU). Figure 1 shows a schematic overview of this architecture. Note that the projector network $p$ has approximately $2048 \cdot 8192 + 2 \cdot 8192^2 = 150{,}994{,}944 \approx 151$ million parameters (the two batch normalization layers have negligible numbers of learnable parameters) whereas the standard ResNet-50 architecture utilizes only 25.6 million parameters.

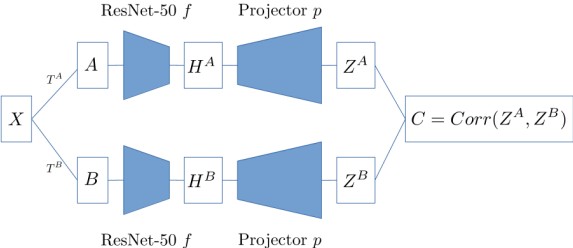

Figure 1: Diagram of Barlow Twins. A batch of images $\boldsymbol{X}$ gets augmented twice. Each augmentation gets propagated through the same ResNet-50 $f$ and projector network $p$. The resulting feature vector matrices $\boldsymbol{Z}^A$ and $\boldsymbol{Z}^B$ are used to calculate $\boldsymbol{C}$.

Propagating a batch through the resulting network can be split up into three steps:

$$\boldsymbol{A} = T^A(\boldsymbol{X}), \quad \boldsymbol{B} = T^B(\boldsymbol{X}),$$
$$\boldsymbol{H}^A = f(\boldsymbol{A}), \quad \boldsymbol{H}^B = f(\boldsymbol{B}),$$
$$\boldsymbol{Z}^A = p(\boldsymbol{H}^A), \quad \boldsymbol{Z}^B = p(\boldsymbol{H}^B).$$

During the first step, the two views are generated by transforming each image in the batch in two ways by $T^A$ and $T^B$ (e.g. color jitter, random grayscale or solarization; for more details see Zbontar et al. (2021)). Note that each image gets transformed by a different augmentation, i.e. $T^A$ for the first image may be different than $T^A$ for the second image. Afterwards, the two views $\boldsymbol{A}$ and $\boldsymbol{B}$ are propagated through the same ResNet-50 $f$. The resulting features $\boldsymbol{H}^A$ and $\boldsymbol{H}^B$ are then propagated through the projector network $p$ to create $\boldsymbol{Z}^A$ and $\boldsymbol{Z}^B$, respectively.

Now, the cross-correlation matrix for the given batch of images is calculated by

$$\boldsymbol{C}_{i,j} = \frac{\langle \boldsymbol{Z}^A_{:,i}, \boldsymbol{Z}^B_{:,j} \rangle}{||\boldsymbol{Z}^A_{:,i}|| \cdot ||\boldsymbol{Z}^B_{:,j}||}. \tag{1}$$

To be precise, it is not the cross-correlation matrix but the matrix of the Pearson correlation coefficients between each pair of output neurons of the projection network. We will come back to this later. The proposed loss function is defined as

$$L_{BT}(\boldsymbol{C}) = \sum_i (1 - \boldsymbol{C}_{i,i})^2 + \lambda \cdot \sum_{i \neq j} \boldsymbol{C}_{i,j}^2 \tag{2}$$

with the cross-correlation matrix $\boldsymbol{C}$. $L_{BT}$ is minimized by the identity matrix, i.e. diagonal components of 1 and off-diagonal components of 0. The regularization parameter $\lambda$ controls the influence of the off-diagonal components.

An intuitive explanation for the loss function is an invariance term for the first sum and a redundancy reduction term for the second sum. The invariance term produces embeddings that are "invariant" to distortions $T^A, T^B$ in the sense that an image feature encoded by an embedding dimension shall highly correlate in different distortions of the same image. Additionally, it avoids the trivial solution of all features being zero. The redundancy reduction term decorrelates different embedding dimensions (features) in order to avoid encoding similar image properties in multiple dimensions. Additionally, it avoids the trivial solution of all features being constant.

After the pretraining phase is concluded, a simple linear classification head is trained via supervised learning. In this phase, the ResNet-50 backbone's parameters are frozen and not further updated, and the projector network is discarded. The classification head is a linear layer that takes in the feature vectors that were extracted by the backbone and assigns a class label. A different interpretation is that the ResNet-50 backbone transforms the images into an embedding space where, in the best case, the features are linearly seperable such that a linear layer can easily classify the respective image.

The Barlow Twins framework has been successfully applied in different scenarios. Bielak et al. (2022) presented the application of the framework for graph neural networks, where the encoder

was pretrained and shown to be on par or better than SOTA models. The authors did not use a projector network, but calculated the cross-correlation matrix directly on the output features. Punn and Agarwal pretrained the encoder of a U-Net (Punn & Agarwal, 2022). After pretraining, the decoder network is initialized with default values. In contrast to the "classic" Barlow Twins framework, the full U-net is finetuned after pretraining on the desired (small) dataset rather than using a frozen encoder network. In Marsocci & Scardapane (2023), the authors extended the Barlow Twins framework for continual learning. In continual learning, large datasets are decomposed into subsets to process them accumulatively. The authors could show that combining self-supervised learning and continual learning leads to great performance by incorporating unlabeled data. Anton et al. (Anton et al., 2023) adapted the Barlow Twins framework in the audio domain. Mohammadamini et al. used the Barlow Twins framework for speaker recognition (Mohammadamini et al., 2022). Their approach used the Barlow Twins objective function as a regularization method while training a ResNet end-to-end on speaker embeddings in both noisy and clean environments.

The mentioned approaches consistently yield results comparable to or surpassing the state-of-the-art, all achieved with a relatively small labeled dataset. In the following, we will demonstrate that this effectiveness of the Barlow Twins approach, in fact, is not achieved by Barlow's redundancy principle but rather by enforcing invariance through the normalization in equation 1. In Garrido et al. (2023) the authors analysed the the connection between sample- and feature-contrastive learning by focussing on the non-diagonal elements of the cross-correlation matrix. Here we focus on the role of the diagonal elements.

## 2 A LINEAR PROJECTOR NETWORK CAN DIAGONALIZE THE CROSS-CORRELATION MATRIX

The Barlow Twins loss equation 2 is not using the precise definition of the cross-correlation matrix, but taking a matrix with the Pearson correlation coefficients between pairs of output neurons of the projector network. However, the Barlow Twins idea of redundancy reduction, i.e., maximizing the correlation between corresponding features and minimizing the correlation between non-corresponding features in distorted views of the same image, can already be achieved with the precise definition of the cross-correlation matrix, which gives

$$\tilde{\boldsymbol{C}}_{i,j} = \langle \boldsymbol{Z}_{:,i}^A, \boldsymbol{Z}_{:,j}^B \rangle \tag{3}$$

without the normalization as in equation 1. Using it in equation 2 also avoids trivial solutions like overall zero or constant values.

For determining $\boldsymbol{C}_{i,j}$ as well as $\tilde{\boldsymbol{C}}_{i,j}$, each image is augmented by $T^A$ and $T^B$. For each image the two augmentations $T^A$ and $T^B$ are drawn independently from the same discrete set of possible augmentations $\mathcal{T}$, hence, each pair of possible $T^A$ and $T^B$ can occur with equal probability. During the training process, the cross-correlation matrix is calculated only over the given batch, a limited number of images. The underlying complete cross-correlation matrix of the Barlow Twins is given by the cross-correlation matrix over the whole data distribution $\mu(\boldsymbol{x})$ and all combinations of $T^A$ and $T^B$. With $\boldsymbol{z}^A(\boldsymbol{x}) = p(f(T^A(\boldsymbol{x})))$ and $\boldsymbol{z}^B(\boldsymbol{x}) = p(f(T^B(\boldsymbol{x})))$ as the output vectors of the projection network for a given $\boldsymbol{x}$ from the data distribution $\mu(\boldsymbol{x})$, the complete cross-correlation matrix 3 is given by

$$\tilde{\boldsymbol{C}}_{i,j} = \int \sum_{T^A, T^B \in \mathcal{T}} \boldsymbol{z}_i^A(\boldsymbol{x}) \boldsymbol{z}_j^B(\boldsymbol{x}) \, d\mu(\boldsymbol{x}). \tag{4}$$

If we take a linear projector network $p$, it can be decribed by a matrix $\boldsymbol{W}$ and we obtain $\boldsymbol{z}^A(\boldsymbol{x}) = \boldsymbol{W} f(T^A(\boldsymbol{x}))$ and $\boldsymbol{z}^B(\boldsymbol{x}) = \boldsymbol{W} f(T^B(\boldsymbol{x}))$ with $f$ as the output vector of the backbone network. It is easy to see that

$$\begin{aligned}
\tilde{\boldsymbol{C}}_{i,j} &= \int \sum_{T^A, T^B} [\boldsymbol{W} f(T^A(\boldsymbol{x}))]_i [\boldsymbol{W} f(T^B(\boldsymbol{x}))]_j \, d\mu(\boldsymbol{x}) \\
&= \int \sum_{T^A, T^B} [\boldsymbol{W} f(T^A(\boldsymbol{x})) f(T^B(\boldsymbol{x}))^T \boldsymbol{W}^T]_{i,j} \, d\mu(\boldsymbol{x})
\end{aligned}$$

and, hence,

$$\tilde{\boldsymbol{C}} = \boldsymbol{W} \boldsymbol{S} \boldsymbol{W}^T \tag{5}$$

with

$$\begin{aligned}
\boldsymbol{S} &= \int \sum_{T^A, T^B \in \mathcal{T}} f(T^A(\boldsymbol{x})) f(T^B(\boldsymbol{x}))^T \ d\mu(\boldsymbol{x}) \\
&= \int \left( \sum_{T \in \mathcal{T}} f(T(\boldsymbol{x})) \right) \left( \sum_{T \in \mathcal{T}} f(T(\boldsymbol{x})) \right)^T \ d\mu(\boldsymbol{x}) \\
&= \int \boldsymbol{y}(\boldsymbol{x}) \boldsymbol{y}(\boldsymbol{x})^T \ d\mu(\boldsymbol{x})
\end{aligned} \tag{6}$$

and

$$\boldsymbol{y}(\boldsymbol{x}) = \sum_{T \in \mathcal{T}} f(T(\boldsymbol{x})).$$

With equation 6, the matrix $\boldsymbol{S}$ is not only symmetric, but also positive semi-definite with non-negative eigenvalues. It is well known from linear algebra that for a symmetric, positive definite matrix $\boldsymbol{S}$ there is always a $\boldsymbol{W}$ in equation 5 that leads to a diagonal $\tilde{\boldsymbol{C}}$ with non-negative diagonal elements. As long as there are no zero diagonal elements (zero eigenvalues), a subsequent linear scaling (whitening) operation leads to $\tilde{\boldsymbol{C}} = \boldsymbol{I}$ with $\boldsymbol{I}$ as the identity matrix which perfectly minimizes the Barlow Twins loss 2. It suffices that the backbone $f$ simply adapts such that the $\boldsymbol{y}(\boldsymbol{x})$ span the whole space to ensure non-zero eigenvalues which leads to perfect solutions for the Barlow Twin loss.

The requirement of non-zero eigenvalues (diagonal elements) is crucial to avoid degenerate solutions such as constant or zero outputs from the backbone. However, this condition alone is insufficient to obtain good solutions, as even random backbone weights can span the entire space and allows even a linear projector head to minimize the Barlow Twins loss perfectly. An MLP projector is far more expressive than a linear layer and can also diagonalize the outputs of random backbones. As a result, using the Barlow Twins loss without normalization of the cross-correlation matrix lacks the necessary formative power for effective backbone pretraining. We now demonstrate how normalization of the cross-correlation matrix changes this dynamic.

## 3 FEATURE NORMALIZATION IS CRUCIAL

The cross-correlation matrix 1 used in the Barlow Twins framework goes beyond simply measuring correlation and decorrelation of features, as already accomplished by the matrix in 3, but also normalizes the correlations. To be precise, it is the Pearson correlation which is used. We demonstrate that this normalization is essential to the success of the Barlow Twins approach. As shown in the previous section, using the raw cross-correlation matrix alone is insufficient for effective backbone pretraining. However, we show that using the Pearson correlation makes this possible.

Analog to equation 4, the complete Barlow-Twins loss is determined by the Pearson correlation over the whole data distribution $\mu(\boldsymbol{x})$ and all combinations of $T^A$ and $T^B$. We obtain

$$\boldsymbol{C}_{i,j} = \frac{\int \sum_{T^A, T^B \in \mathcal{T}} \boldsymbol{z}_i^A(\boldsymbol{x}) \boldsymbol{z}_j^B(\boldsymbol{x}) \ d\mu(\boldsymbol{x})}{\sigma_i \sigma_j} \tag{7}$$

with

$$\sigma_i = \sqrt{\int \sum_{T \in \mathcal{T}} [\boldsymbol{z}_i(\boldsymbol{x})]^2 \ d\mu(\boldsymbol{x})}$$

as the standard deviation of the $i$-th output neuron of the projector network over all $\boldsymbol{x}$ from $\mu(\boldsymbol{x})$ and all augmentations $T \in \mathcal{T}$. Accordingly, $\sigma_j$ is the standard deviation of the $j$-th output neuron. Note that $\boldsymbol{C}_{i,j}$ is symmetric.

The Pearson correlation ranges between $-1$ and $+1$, with $+1$ indicating perfect positive correlation. In the optimum of the Barlow Twins loss, this perfect positive correlation is required for the diagonal elements $\boldsymbol{C}_{i,i}$. $\boldsymbol{z}_i^A(\boldsymbol{x})$ and $\boldsymbol{z}_i^B(\boldsymbol{x})$ correlate perfectly positively, if and only if for each $\boldsymbol{x}$

$$\boldsymbol{z}_i^A(\boldsymbol{x}) = a \boldsymbol{z}_i^B(\boldsymbol{x}) + b \qquad \text{for each pair } T^A, T^B \in \mathcal{T}$$

is valid for a fixed $a, b \in \mathbb{R}$ with $a > 0$. However, this implies that for a given $\boldsymbol{x}$, this equation must be valid for a $T^A, T^B$ as well as vice versa for $T^B, T^A$. Then, for the given $\boldsymbol{x}$ we obtain $\boldsymbol{z}_i^A(\boldsymbol{x}) = a\boldsymbol{z}_i^B(\boldsymbol{x}) + b$ and $\boldsymbol{z}_i^B(\boldsymbol{x}) = a\boldsymbol{z}_i^A(\boldsymbol{x}) + b$. Subtracting both equations yields $\boldsymbol{z}_i^A(\boldsymbol{x}) - \boldsymbol{z}_i^B(\boldsymbol{x}) = a(\boldsymbol{z}_i^B(\boldsymbol{x}) - \boldsymbol{z}_i^A(\boldsymbol{x}))$. Since $a > 0$, this is valid if and only if $\boldsymbol{z}_i^A(\boldsymbol{x}) = \boldsymbol{z}_i^B(\boldsymbol{x})$.

Since this holds for each $i$ and $T^A, T^B$, we can conclude that in the minimum of the Barlow-Twins loss with the Pearson correlation for any image $\boldsymbol{x}$ the corresponding output vector $\boldsymbol{z}(\boldsymbol{x})$ must remain invariant to any distortion (augmentation) of the image $\boldsymbol{x}$. This is equivalent to the minimum of

$$\int \sum_{T^A, T^B \in \mathcal{T}} ||\boldsymbol{z}^A(\boldsymbol{x}) - \boldsymbol{z}^B(\boldsymbol{x})||^2 \, d\mu(\boldsymbol{x}) \tag{8}$$

in sample contrastive learning approaches.

## 4 CONCLUSION

We demonstrated that the core idea behind Barlow Twins — reducing redundancy by maximizing the correlation between corresponding features and minimizing the correlation between non-corresponding features in distorted images — is insufficient on its own for pretraining a backbone network. The cross-correlation matrix of arbitrary outputs, even from untrained backbones, is symmetric and, hence, can be diagonalized already by a linear projection head. A multilayer perceptron (MLP) as a projection head is even more expressive at achieving diagonalization than a simple linear layer. However, since non-zero eigenvalues of the cross-correlation matrix are enforced, it is ensured that the backbone output spans the entire space, preventing degenerate solutions such as constant or zero output. This is one of the conditions for which usually explicit terms in the cost function are used. For example, VICReg (Bardes et al., 2022) employs a term in the cost function which enforces non-zero variance.

The primary objective of self-supervised learning is to achieve invariance in the backbone output for perturbed images. Approaches like VICReg and other contrastive methods explicitly incorporate terms like 8 in the cost function. We showed that the Barlow Twins method achieves the same invariance by using the Pearson correlation matrix, which inherently includes normalization. This normalization is crucial, as it enforces the same minima in the cost function as the explicit term 8. Without this normalization, the Barlow Twins loss would not yield meaningful solutions.

In conclusion, Barlow Twins offer an elegant approach to achieving the key objectives of self-supervised learning: They ensure invariance in the network output to image perturbations like explicit terms used in contrastive approaches, while simultaneously avoiding trivial solutions like constant or zero output, by simply diagonalizing the Pearson correlation matrix with non-zero diagonal elements.

### ETHICS STATEMENT

This paper presents rather theoretical work whose goal is to advance the understanding of the principles of Machine Learning. There are many potential societal consequences of our work, none of which we feel must be specifically highlighted here.

### REPRODUCIBITILY STATEMENT

As a purely theoretical paper, all mathematical derivations are fully detailed, with every step clearly outlined. Any necessary assumptions are explicitly stated.

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
