# OpenReview forum: "Why Barlow Twins Work: The Critical Role of Normalization and Its Link to Sample Contrastive Learning"
_ICLR.cc/2025/Conference — ICLR 2025 Conference Withdrawn Submission_

### Official Review · Reviewer_4jJS · 2024-10-29

**Soundness:** 1
**Presentation:** 1
**Contribution:** 2
**Rating:** 1
**Confidence:** 5

**Summary:**

This paper analyzes the mechanism of Barlow Twins from a new theoretical perspective, highlighting the importance of the normalization of the cross-correlation matrix. The authors prove that only the invariance and redundancy term in the Barlow Twins objective is insufficient to obtain meaningful representations.

**Strengths:**

The theoretical analysis of the underlying mechanism behind feature-contrastive methods is important and still under-explored. The perspective in this paper is interesting.

**Weaknesses:**

1. In this paper, the authors do not provide empirical evidence for the analysis.
2. The theoretical analysis in Section 3 is quite confusing. The analysis does not explain why the feature normalization is crucial and how it influences the dynamic of the Barlow Twins objective.
3. It seems that the paper is unfinished. I think 5 pages is too short to establish a new theoretical analysis of the Barlow Twins method and the authors do not prove the effectiveness of the theoretical framework.
4. In this paper, the authors do not deliver insights about how can we further improve the Barlow Twins or current self-supervised methods.

**Questions:**

See the weaknesses.

---

### Official Review · Reviewer_ui9a · 2024-11-01

**Soundness:** 1
**Presentation:** 1
**Contribution:** 1
**Rating:** 1
**Confidence:** 4

**Summary:**

This paper shows it is important to use the normalization for calculating Pearson correlation matrix in Barlow Twins. This normalization enforces the same minima in the cost function as the explicit term. Without this normalization, the Barlow Twins loss would not yield meaningful solutions.

**Strengths:**

The paper is well-written and it clearly explains the proposed method.

**Weaknesses:**

Missing experiments for supporting the proposed demonstration.

**Questions:**

NA

---

### Official Review · Reviewer_GbdD · 2024-11-03

**Soundness:** 3
**Presentation:** 3
**Contribution:** 1
**Rating:** 1
**Confidence:** 5

**Summary:**

This paper provides an additional theoretical analysis of the Barlow Twins (BT) framework for self-supervised learning (SSL); wherein the authors higlight that a key contributor behind the efficacy of Barlow Twins is the normalization as part of the cross-correlation matrix diagonalization process which avoids the collapse into degenerate solutions during training.

**Strengths:**

The paper itself is quite well written, and fairly easy to follow. Moreover, I do find the insight that BT does not work when simply used with non-normalized cross-correlation to be interesting.

**Weaknesses:**

Unfortunately, this paper feels very incomplete, and has several severe shortcomings that in my eyes stop it from being a somewhat complete scientific study.

First and foremost, the paper itself latches onto differences in formulation; wherein the original BT paper uses the cross-correlation matrix of pearson correlation coefficients but simply dubs it to be a "cross-correlation" matrix.
The authors then provide a deeper dive into why the use of a simple cross-correlation matrix is theoretically insufficient to learn meaningful representations; as the desired diagonalization in BT of a simple cross-correlation matrix can be solved through simple shortcuts; before then showcasing why the normalization used in BT solves this issue.

To me, this is tackling and addressing a problem that is simply not there. BT is an established method, and as even the authors note, in its explicit setup sound. It is merely a slight inaccuracy in definition. While interesting, the correspondingly provided insight have in my eyes no meaningful relevance to practitioners in SSL, as it tackles a non-existent issue.

And even when disregarding this fundamental aspect, the paper itself is simply insufficient:

* Roughly 50% of this paper has no novelty at all; being made up by abstract, introduction and description of Barlow twins.

* There is no experimental study in which the authors provide case study support on the break-down of Barlow twins without the normalization component; e.g. how likely is it for the model to arrive at degenerate solutions in practice? And are there other ways beyond simple normalization that may be explored alongside a standard cross-correlation formulation?

* Is there any impoact on the multitude of other SSL objectives that are now widely established and have in a large number of cases already replaced BT (s.a. VicReg variations)?

**Questions:**

See Weaknesses. I am currently strongly advocating for rejection, as the submission provides insufficient (no) experimental support, and a theoretical breakdown of a non-existent problem. I am only willing to raise my score if the authors can provide convincing arguments as to why their theoretical study on the hypothetical differentiation between cross-correlation and the normalized variant used in BT is needed; alongside some experimental studies in support.

---

### Official Review · Reviewer_kiMZ · 2024-11-04

**Soundness:** 2
**Presentation:** 2
**Contribution:** 1
**Rating:** 3
**Confidence:** 4

**Summary:**

The authors argue that the success of Barlow Twins comes from the normalization of the cross-correlation matrix rather than other parst of the criterion. To do so, they show that it is trivial to minimize the off-diagonal terms of the cross-correlation matrix (indeed, random Gaussian embeddings minimize it). Instead, the normalization of the embeddings is key to obtain the same minimizer as contrastive methods, and thus learn informative representations.

**Strengths:**

- The paper proposes to analyze theoretically the importance of normalization in Barlow Twins. While present in almost all SSL methods, normalization is often left as an implementation detail, so trying to understand its impact is a welcome goal.

**Weaknesses:**

- The proof that the cross correlation is symmetric (Equation 6) hinges on a few assumptions that are most likely unrealistic during training. Mainly, here, all possible pairs of augmented images are used to compute C (or S equivalently). In practice, we only see a unique pair, with $T^A \neq T^B$ almost surely. In this setting, unless the invariance part of the criterion is satisfied, we will not obtain an empirical cross correlation matrix that is symmetric. This may ensure that no trivial solutions as discussed in section 2 will arise in practice.

- Equation (8) is an obvious result of the property $Tr(AB) = Tr (BA)$. This implies that the sum of terms in the diagonal of the cross correlation matrix is the same as the sum of terms in the diagonal of the gram matrix. The latter is simply the sum of dot products between pairs of augmented images (and thus how similar they are). This result is independent of the normalization strategy used. The normalization is then intimately tied to the target of 1 in the barlow twin loss to help maximize the similarity between augmented pairs.

- A common intuition as to why normalization is necessary is that it prevents blowing up or reducing the norm of embeddings to reduce the loss. It would thus be here more to avoid degenerate/unstable solutions rather than be the key to learning meaningful features. The discussions on normalization in [1] (Lemma 3.4, corollary 3.4.1, table 1) would also suggest that any kind of normalization is good enough, similarly to table 5 in Barlow Twins’ paper.  The conclusion that “Without this normalization, the Barlow Twins loss would not yield meaningful solutions” (l 247) should thus be more nuanced.

[1] Garrido, Q., Chen, Y., Bardes, A., Najman, L., & Lecun, Y.. On the duality between contrastive and non-contrastive self-supervised learning. ICLR,2023.

**Questions:**

- Line 186-187 “ An MLP projector is far more expressive than a linear layer and can also diagonalize the outputs of random backbone”. Is there a reference/proof for this result ?

- It is argued that using the raw cross-correlation matrix is insufficient for training (e.g. lines 184-186 and 197-198). However, looking at table 5 in the Barlow Twins paper, we can see that this scenario still achieves 53.7% accuracy on ImageNet, which while low, is far from trivial. Is there a way to reconcile this empirical result with the arguments from section 2 ?


Overall, I feel that the ablations from the Barlow Twins paper already show the importance of normalization strategy. The theoretical results discussed in the paper also do not seem convincing enough on the importance of the normalization strategy, which would shed more light on the empirical evidence. Bad minimizers to the off-diagonal part of the loss are also known to exist (e.g. using random vectors with entries i.i.d. from N(0,1)), and the link between invariance to augmentation and diagonal terms of the cross-correlation matrix is rather trivial (it is also discussed intuitively in the introduction of the Barlow Twins’ paper).

---

### Note · Authors · 2024-11-22

I have read and agree with the venue's withdrawal policy on behalf of myself and my co-authors.